# Effect of Eco-Friendly Application of Bee Honey Solution on Yield, Physio-Chemical, Antioxidants, and Enzyme Gene Expressions in Excessive Nitrogen-Stressed Common Bean (*Phaseolus vulgaris* L.) Plants

**DOI:** 10.3390/plants12193435

**Published:** 2023-09-29

**Authors:** Hussein E. E. Belal, Mostafa A. M. Abdelpary, El-Sayed M. Desoky, Esmat F. Ali, Najla Amin T. Al Kashgry, Mostafa M. Rady, Wael M. Semida, Amr E. M. Mahmoud, Ali A. S. Sayed

**Affiliations:** 1Botany Department, Faculty of Agriculture, Fayoum University, Fayoum 63514, Egypt; hes00@fayoum.edu.eg (H.E.E.B.); mabdelpary2013@gmail.com (M.A.M.A.); 2Botany Department, Faculty of Agriculture, Zagazig University, Zagazig 44519, Egypt; sayed1981@zu.edu.eg; 3Department of Biology, College of Science, Taif University, P.O. Box 11099, Taif 21944, Saudi Arabia; a.esmat@tu.edu.sa (E.F.A.); n.takth@tu.edu.sa (N.A.T.A.K.); 4Horticulture Department, Faculty of Agriculture, Fayoum University, Fayoum 63514, Egypt; wms00@fayoum.edu.eg; 5Biochemistry Department, Faculty of Agriculture, Fayoum University, Fayoum 63514, Egypt; aem01@fayoum.edu.eg

**Keywords:** beans, nutrient toxicity, plant biostimulators, production quality, transcription levels

## Abstract

Excessive use of nitrogen (N) pollutes the environment and causes greenhouse gas emissions; however, the application of eco-friendly plant biostimulators (BSs) can overcome these issues. Therefore, this paper aimed to explore the role of diluted bee honey solution (DHS) in attenuating the adverse impacts of N toxicity on *Phaseolus vulgaris* growth, yield quality, physio-chemical properties, and defense systems. For this purpose, the soil was fertilized with 100, 125, and 150% of the recommended N dose (RND), and the plants were sprayed with 1.5% DHS. Trials were arranged in a two-factor split-plot design (N levels occupied main plots × DH– occupied subplots). Excess N (150% RND) caused a significant decline in plant growth, yield quality, photosynthesis, and antioxidants, while significantly increasing oxidants and oxidative damage [hydrogen peroxide (H_2_O_2_), superoxide (O_2_^•−^), nitrate, electrolyte leakage (EL), and malondialdehyde (MDA) levels]. However, DHS significantly improved antioxidant activities (glutathione and nitrate reductases, catalase, ascorbate peroxidase, superoxide dismutase, proline, ascorbate, α-tocopherol, and glutathione) and osmoregulatory levels (soluble protein, glycine betaine, and soluble sugars). Enzyme gene expressions showed the same trend as enzyme activities. Additionally, H_2_O_2_, O_2_^•−^, EL, MDA, and nitrate levels were significantly declined, reflecting enhanced growth, yield, fruit quality, and photosynthetic efficiency. The results demonstrate that DHS can be used as an eco-friendly approach to overcome the harmful impacts of N toxicity on *P. vulgaris* plants.

## 1. Introduction

Common bean (*Phaseolus vulgaris* L.) is a global widely consumed leguminous crop rich in minerals, vitamins, carbohydrates, and proteins essential for human nutrition. In addition, it serves as a crucial source of income for small farmers in developing nations [1]. Nevertheless, various environmental stresses, such as excessive N use, often hinder crop production [2].

In contemporary farming, normal plant development and growth require N as a crucial nutrient. As a result, N-based fertilizers are extensively utilized to enhance crop yield and overall quality. Efficient utilization of N is of utmost importance to sustainably provide food for the world’s population [2]. Despite this fact, farmers in numerous regions around the world tend to overuse N to enhance crop productivity [3]. However, overuse of N fertilizers can deteriorate the soil environment [4] and pollute groundwater, causing many environmental and human hazards [5,6,7]. It can promote pathogen development, reduce root growth and nutrient uptake, photosynthetic efficiency, and crop yields [8].

N toxicity in plants engenders several harmful mechanisms such as the production of high contents of nitrate (NO_3_^−^) and reactive oxygen species (ROS) [9], including hydrogen peroxide (H_2_O_2_) and superoxide (O_2_^•−^). The accumulation of NO_3_^−^ is extremely harmful to human health [10]. The increased levels of ROS due to excessive N use can damage many cellular components (e.g., lipids, DNA, proteins, etc.), resulting in decreased membrane stability and degradation of chlorophyll, ultimately leading to oxidative damage and cellular dysfunction [11]. Additionally, excessive N use adversely affects photosynthetic efficiency, nutrient and hormonal homeostasis, osmoregulation, and different enzymatic and non-enzymatic antioxidant systems [10]. Therefore, plants have evolved complex systems implicated in defense and overcoming oxidative damage. These systems include enzymatic antioxidants (e.g., glutathione reductase, GR; superoxide dismutase, SOD; catalase, CAT; ascorbate peroxidase, APX; etc.), antioxidant compounds (ascorbate, AsA; glutathione, GSH; proline; alkaloids; flavonoids; carotenoids; α-tocopherol; phenolics; etc.) [12,13], and osmoregulatory compounds (ORCs) like simple sugars, soluble proteins, and glycine betaine [14,15,16,17,18].

Biostimulators (BSs) refer to products derived from a variety of organic or inorganic materials and/or microorganisms, which can induce processes to promote nutrient uptake, utilization efficiency, plant growth, and yield quality. They also mitigate the harmful impacts of abiotic stressors when used in small amounts [19,20,21,22]. They can positively change plant physiology, primary and secondary metabolism, and certain molecular functions that help crops to use water and nutrients effectively, resulting in improved plant growth and coping with abiotic stresses [23,24]. Biostimulators are extracts that have stimulatory effects on plants, mainly signalizing a hormonal mode of action [19]. They provide numerous growth-promoting mechanisms, including delaying senescence (physiological modification), increasing micronutrients (biochemical change), improving water use efficiency due to better regulation of differential genes, and increasing activities of rhizobacteria and mycorrhizae (rhizosphere impacts) [19,23,24].

A diluted bee honey solution (DHS) contains BSs, including ORCs, nutrients, vitamins, antioxidants, flavonoids, phenolic, inorganic, and organic acids, which effectively enhance tolerance to plant stress impacts [17,24,25,26]. These BSs can enter the cells of plant leaves upon application, and participate as key mechanisms responsible for direct improvements in plant growth. Furthermore, DHS exhibits remarkable efficacy in eliminating 2,2-diphenyl-1-picrylhydrazyl (DPPH) radicals and has been shown to elicit physiological, biochemical, and antioxidant alterations that suppress ROS levels and boost stress tolerance [17,25,26,27]. Additionally, the enzymes and flavonoids in DHS hinder the peroxidation process and play a role in removing ROS [27]. Under different stresses, DHS application suppresses oxidative stress biomarker levels and oxidative damage, resulting in improved plant growth, yield, output quality, photosynthesis regulation, nutrient and hormonal balances, leaf integrity, ORCs levels, antioxidants and enzyme activities, and enzyme gene expressions [17,25].

No research has studied the potential beneficial effects of DHS to mitigate excessive N-induced toxic stress in plants at the field level. Based on previous findings that DHS mitigated the deleterious impacts in salinity-stressed onion plants, water undersupply stressed faba bean plants, and saline-calcareous-stressed Atriplex seedlings [17,24,25], this study hypothesized that foliar nourishment with DHS would mitigate the toxic influences of N overuse in *P. vulgaris* plants, reflecting improvements in growth, yield, and yield quality while minimizing contamination (nitrate content in plant edible parts). Therefore, the aim of this investigation was to study the use of DHS (as BSs) to attenuate the detrimental influences of N toxicity stress on growth, physio-chemical properties, antioxidant defense systems, enzyme gene expression, yield, and quality of the edible part of *P. vulgaris* L. plants.

## 2. Materials and Methods

### 2.1. Trial Location and Soil Analyses

An area of 700 m^2^ was identified on a private farm (Bani Suef District; latitude of 29.064337° N, longitude 31.020612° E, Egypt) with clay loam soil to conduct a field study for two autumns (2021 and 2022). Appendix A shows the average climate observations for the two seasons in the Bani Suef district. The tested soil was analyzed to evaluate its physicochemical properties [28,29]. Appendix A shows the soil analysis observations, indicating that it is normal (EC_e_ = 2.34 dS m^−1^) [30].

### 2.2. Planting, Treatments, and Experimental Layout

The Egyptian Agricultural Research Center provided *Phaseolus vulgaris* seeds (cultivar Bronco) for this study. A 1% (*v*/*v*) solution of NaClO was prepared to disinfect the seeds for 2 min, distilled water was utilized to clean them thoroughly to eliminate the sterilization solution, and then left to air-dry at 22 ± 2 °C. On September 10, in both autumns (2021 and 2022), the seeds were planted in hills spaced 20 cm apart in each row; they were 50 cm wide and 3.5 m long, with each 6 rows representing a plot of 10.5 m^2^.

All experiments were performed by applying a split-plot arrangement with two major factors (3 N levels applied in main 3 soil strips as 100, 125, and 150% of the recommended dose × 2 DHS concentrations; 0 and 1.5%, which were applied in subplots). The DHS treatments were scattered in each N application split in a CRD (completely randomized design) with 6 replications. The first soil strip was treated with 300 kg ammonium nitrate (33% N) per hectare as the recommended N dose (RND). The second and third strips were treated with 375 kg (125% RND) and 450 kg (150% RND) per hectare. Half of these rates were applied while preparing the soil for planting. The second half was supplied to the soil immediately before the second irrigation. The N fertilizer was spread manually on the soil surface. In addition, 200 kg K_2_SO_4_ (50% K_2_O) + 200 kg CaH_6_O_9_P_2_ (12% P_2_O_5_) were added to the soil in two portions as carried out with N. The three soil strips were separated by 2 m boundaries to prevent overlapping of N treatments. After 3, 4, and 5 weeks of planting, plants were sprayed with 0% DHS (distilled water as control) and 1.5% DHS. The DHS was prepared by dissolving each 15 mL of fresh clover bee honey in a liter of distilled water immediately before the spraying. For optimal penetration into leafy tissue, the spray solutions were enriched with a 0.1% surfactant (Tween-20). Within each strip of soil, the plots were separated by 1 m boundaries to prevent overlapping of DHS treatments. Table 1 presents the analysis (physicochemical properties) of raw bee honey samples. The honey has a lower pH (4.02) due to its higher content of organic acids. It also has higher contents of proteins, different macro and micronutrients, including iodine and selenium, various antioxidants, including vitamin C and B-group vitamins, and osmoregulatory compounds (ORCs) like proline and many soluble simple sugars. The valuable BSs of DHS endow it with high DPPH-radical scavenging activity (89.5%). Thus, utilizing 1.5% DHS can act as an efficient eco-friendly approach to mitigate the detrimental impacts of excessive N application on *P. vulgaris* plants. All the raw honey’s physicochemical properties were determined by applying the procedures depicted in A.O.A.C. [31].

Irrigation was performed early in the morning. It was applied with a total water volume of 6650 m^3^ ha^−1^. A total water volume of 950 m^3^ ha^−1^ was applied every 10 days. Appendix A includes the major composition of irrigation water. Based on the recommendations of the Egyptian Center for Agricultural Research, weeds were controlled via manual hoeing once a week. In addition, the pathogens were controlled biologically using biocides; Bio-Arc (6.0% *Bacillus megaterium*; 2.5 × 10^7^ cells/g) and Bio-Zeid (2.5% *Trichoderma album*; 10 × 10^6^ spores/g). Each biocide was added at 2.5 g for each per kg of seeds.

### 2.3. Plant Sampling

Forty-five days after seeding (DAS), a group of 18 plants (3 plants from the middle two rows for each of the 6 replicates) in each treatment were selected randomly and sampled. Two more groups of 18 plants were selected randomly in each treatment in the same way and sampled. The first group of plant samples was allocated to determine growth traits, photosynthesis efficiency, and nutrient contents. The second group was allocated to examine enzyme activities, enzyme gene expressions, and hormonal analysis. The third group was recruited to determine physiological features, oxidative stress markers, non-enzymatic antioxidants, ORCs, and nitrate levels. Starting at 45 DAS, pods were collected six times at three-day intervals, from plants in the other 4 rows in each plot. The pods were used to evaluate green pod yield and quality components.

### 2.4. Evaluation of Growth, Green Pod Yield, and Pod Quality

Leaves per plant were counted, and the leaf area was evaluated utilizing a planimeter (Digital Planix 7). For dry weight, the shoots were dried at 70 ± 2 °C until the weights reached stability. The pods harvested from the *P. vulgaris* plants were subjected to an evaluation of an average number of pods, pod weight per plant, and total pod yield (tons per ha).

A micro-Kjeldahl device (NM-Instruments Co., Ningbo, China) was utilized to evaluate the percentage of protein in the pods by determining the percentage of total N (N% × 6.25 = pod protein percentage) [31]. The methods of Sadasivam and Manickam [32] and Rai and Mudgal [33] were utilized to evaluate the percentages of carbohydrates and fiber, respectively, in the pods. After digestion of the dried pods with a mixture of 1 HClO_4_: 3 HNO_3_ (*v*/*v*), the percentages of P, N, and K were determined [28,31,34], respectively. The aforementioned method of micro-Kjeldahl apparatus was applied for the evaluation of N. The P content was evaluated depending on the reduction rate of H_3_PMo_12_O_40_ in H_2_SO_4_ by molybdenum to eliminate arsenic. The content of K^+^ was determined using a flame photometer (Perkin-Elmer Model 52-A, Glenbrook, Stamford, CT, USA). The content of nitrate (NO_3_^−^, mg per g DW) and the activity of NO_3_^−^ reductase (NR) (nM NO_2_^−^ per mg protein per min) were estimated in the pods. The activity of NR was assayed by exploring the accumulation of nitrite (NO_2_^−^) as the reaction product [35]. On the ice, the just collected green pods were disrupted in a Na-P buffer (0.05 M, pH 8.0). The homogenates were centrifuged at 4000× *g* for 15 min at a temperature below 4 °C. Each sample’s resulting supernatant received 100 mM KNO_3_ and 5 mM NADH. Instead of NADH, the reference sample received the same amount of distilled water. After 30 min, the reaction termination was conducted by adding glacial CH_3_COOH. Ten min centrifugation was performed at 8000× *g* to precipitate proteins. The supernatant received a reagent (1% Griess in 12% CH_3_COOH), and absorbance was taken at 527 nm after 30 min. For NO_3_^−^, after drying the pod samples at 70 °C for 72 h, they were ground, and then NO_3_^−^ analysis was performed [36]. The method relies on extracting dry pod NO_3_^−^-N in water at 45 °C for 60 min. After centrifugation at 5000× *g* rpm and filtering, the filtrates (0.5 mL) received a 1 mL solution (5%) of salicylic + sulfuric acids (*w*/*v*) and then a 9.5 mL solution of NaOH (4 N) after 20 min. Using a spectrophotometer (Shimadzu UV-160 A, SpectraLab Scientific Inc., Markham, ON, Canada), the absorbance of NO_3_^−^-N was taken at 410 nm, and KNO_3_ was used as a reference. To obtain the pod content of NO_3_^−^, the coefficient of 4.42 was used.

### 2.5. Estimation of Photosynthesis Efficiency and Leaf Contents of Nutrients

The Arnon [37] method was followed to estimate total chlorophyll (Chl) and carotenoid (Carts) contents (mg per g FW) in the fresh leaf tissue samples. Chlorophyll and Carts were extracted with homogenization of a 100 mg tissue in a 10 mL solution of 80% acetone. For 20 min, the homogenates were set on a centrifuge at 3000× *g*. Overnight, the samples were stored, and then the supernatant absorbance was taken at 480 nm for Carts, and at 645 and 663 nm for Chl. A PEA Chl-fluorometer (Hansatech Instruments Ltd., Kings Lynn, UK) was utilized to estimate Chl-fluorescence. Fv/Fm = (Fm − F0)/Fm was the formula to estimate Fv/Fm [38]. The performance index (PI) of PSII was calculated [39].

As was performed with pods (Section 2.4), P, N, and K contents (mg per g DW) were determined [28,31,34], respectively. The methods in [40] were utilized to evaluate Zn, Mn, and Fe contents (µg per g DW), against NIST (USA) standard reference samples using atomic absorption spectroscopy.

### 2.6. Assessment of Leaf Tissue Integrity and Biomarkers of Oxidative Stress

Relative water content (RWC) was estimated in midrib-free leaf tissues following the procedures depicted in [41]. The estimation of RWC relies on recording the weights of fresh, turgid, and dried leaf discs (FW, TW, and DW, respectively).

Membrane stability index (MSI) was estimated in midrib-free leaf tissues following the procedures depicted in [42]. The estimation of MSI relies on recording the EC (electrical conductivity) of the tissue solution when it is warm (40 °C) (EC_1_) and when it is boiling (100 °C) (EC_2_).

Electrolyte leakage (EL) was estimated in midrib-free leaf tissues following the procedures depicted in [43]. The estimation of EL (the ions leaked out of the leaf tissues to the solution) relies on recording the EC of the tissue solution when it is normal (EC_0_), warm (45–55 °C) (EC_1_), and boiling (100 °C) (EC_2_).

The following formulas were applied to compute RWC, MSI, and EL, respectively:RWC (%) = 100 × [(FW − DW)/(TW − DW)]
MSI (%) = 100 × [1 − (EC_1_/EC_2_)]
EL (%) = 100 × [(EC_1_ − EC_0_)/EC_2_]

Hydrogen peroxide (H_2_O_2_), superoxide (O_2_^•−^), and malondialdehyde (MDA) contents (µM g^−1^ FW) were determined spectrophotometrically [44,45,46], respectively. The content of H_2_O_2_ (0.28 µM^−1^ cm^−1^ as a molar extinction coefficient) was estimated using a weight of 0.5 g of the leaf tissue, which was extracted in acetone. Thereafter, the acetone extract was provided with a titanium (Ti) reagent and NH_4_^+^ and then dissolved in 1 M H_2_SO_4_. The supernatant absorbance was read at 415 nm. The content of O_2_^•−^ was estimated using leaf fragments (1 × 1 mm, 0.1 g) immersed in 10 mM buffer (K-P, pH 7.8) mixed with NaN_3_ (10 mM) + NBT (0.05%). For 1 h, the mixture was stored at 25 °C and then for 15 min at 85 °C. The mixture was speedily cooled, and the absorbance was taken at 580 nm. Applying the method of the thiobarbituric acid (TBA) reaction, MDA content was estimated using a 0.5 g FW sample. The sample was homogenized in 5% C_2_HCl_3_O_2_ and then centrifuged for 10 min at 4000× *g*. Just before boiling for 10 min, 2 mL of TBA (0.6%) was mixed with 2 mL of extract. Then, 532, 600, and 450 nm were applied to record the absorbances, and MDA content was calculated as follows:MDA content = [(A_532_ − A_600_) × 6.45] − [A_450_ × 0.56]

### 2.7. Assessment of Nitrate (NO_3_^−^) Content, Nitrate Reductase (NR) Activity, Osmoregulatory Compounds and Antioxidants

The determinations of NO_3_^−^ content (mg per g DW) and NR activity (nM NO_2_^−^ per mg protein per min) were conducted in leaf samples as were conducted in pods [35,36]; Section 2.4. Soluble sugar content (mg per g DW), as well as proline, ascorbate (AsA), glutathione (GSH), and α-tocopherol (αToC) contents (µM per g DW) were estimated following the procedures of Irigoyen et al. [47], Bates et al. [48], Huang et al. [49], Paradiso et al. [50], and Nagy et al. [51], respectively.

Total soluble sugar content was estimated in the supernatant after centrifugation of the extract, which was gathered to mix with anthrone (freshly prepared). The mixture was incubated for 10 min at 100 °C. The absorbance was measured at 625 nm, and the total soluble sugar content was computed using standard curves prepared with glucose. The content of proline was assessed in the supernatant resulting from the extract centrifugation and mixed with an acid-ninhydrin (freshly prepared solution). After incubation at 90 °C for 0.5 h, the reaction was terminated in ice. Again, toluene was utilized for a second extraction, and the toluene and aqueous phases were separated. The absorbance was read at 520 nm, and standard curves prepared with proline were applied. To extract AsA, leaf tissue was homogenized in a 5% HPO_3_ (ice-cold) solution, containing 1 mM EDTA. The homogenate was subjected to a 20 min centrifugation at 4000× *g*. The supernatant was utilized to estimate AsA content. The content of GSH was estimated, and a standard curve of GSH was utilized. A UV–160A UV–vis spectrometer (Shimadzu, Kyoto, Japan) was applied for the above determinations. The content of αToC was obtained after extracting a leaf sample in a 900 mL solvent containing C_10_H_22_O_2_, C_6_H_14_, and C_4_H_8_O_2_, as well as C_15_H_24_O. With R-Toc, a stock solution was prepared for several standards (20–200 µg mL^−1^). Sample preparation and saponification were performed. Thereafter, dried leaf tissue slices were homogenized and suspended in water, and then enriched with 21 g KOH dissolved in 100 mL of C_2_H_5_OH. Ascorbate (0.25 g) was added for the test. Also, 9 mL N-hexane and 1 mL C_4_H_8_O_2_ were added. Three-time extraction was carried out for the mixture. The organic phases were filtered and evaporated to dryness. The content of αToc was estimated via a mobile phase by the HPLC system.

### 2.8. Assay of Antioxidant Enzyme Activities, Enzyme Gene Expression, and Hormonal Content

Forty-five DAS, for three replicates per treatment, and leaf blades devoid of the midrib were sampled from the third youngest fully developed leaf, and directly subjected to obtain enzyme extracts (EEs); the extractions took place in an ice bath. The EEs were obtained after homogenization of a freeze-dried leaf sample (0.2 g) in 2 mL of 100 mM buffer (K-P, pH 7.0) receiving a 100 mM solution of EDTA. Two mM AsA was added only for assaying APX activity. The filtrate, obtained after filtering, was centrifuged at 12,000× *g* for 15 min. The EEs were applied immediately; otherwise, they were stored under −25 °C. All the above steps were implemented under 4 °C.

The ability of each one-unit activity of superoxide dismutase (SOD) to inhibit 50% photo-reduction in NBT is the basis of assaying SOD activity (Unit g^−1^ protein) [52]. Measuring the degradation of H_2_O_2_ at 240 nm is the basis of quantifying catalase (CAT) activity when H_2_O_2_ is added to EEs [53]. With the same basis, 2 mM ascorbate was added to EEs for the activity of ascorbate peroxidase (APX) [54]. As depicted in [55], glutathione reductase (GR) activity was assayed by observing the absorbance at 340 nm for 3 min in the EEs (0.1 mL), which received K-P buffer (0.1 M, pH 7.0), 500 μM NADPH, 100 μM EDTA, and 100 μM GSSG. The activity expression of CAT, APX, and GR was µM H_2_O_2_ per min per g protein. As depicted in [56], the content (mg per g DW) of soluble protein was estimated.

Using an RNeasy Mini Kit (Qiagen GmbH, Hilden, Germany), a leaf sample of *P. vulgaris* was used to isolate total RNA. Then, using a RevertAid H Minus First Strand cDNA Synthesis Kit (Fermentas GmbH, St. Leon-Rot, Germany), the upcoming cDNA was biosynthesized. Appendix A presents the sequences of primers for RT-PCR of genes tested in *P. vulgaris*. For analyzing qRT–PCR, the manufacturer’s directives of iQ SYBR Green Supermix (Bio-Rad, Hercules, CA, USA) were practiced. Two actin reference genes were utilized for the normalization of data. The efficiency of the reactions was calculated with LinRegPCR Software (version 11.0, download: http://LinRegPCR.HFRC.nl, accessed on 5 April 2023) [57]. In the analysis, qPCR raw data were utilized. Baseline corrections were conducted with the baseline trend depending on early cycle choices or using the developed algorithm. For every single sample, PCR efficiency was derived from the slope of the regression line fitted to a subset of baseline-corrected data points in the log-linear phase using LinRegPCR. Posteriorly, the formula of [58] was applied. With simple mathematics, gene expression ratios can be calculated (ratio = N_0,target/_N_0,reference_), besides fold-difference in gene expression among conditions (fold = ratio_experiment/_ratio_control_).

The GC–MS system (Perkin Elmer, Waltham, MA, USA) was applied to estimate IAA, GA_3_, and CK profiling [59,60]. After leaf sample extraction in an ice-cold mixture (80% CH_3_OH: 19.9% H_2_O: 0.1% HCl, *v*/*v*/*v*), the extract was centrifuged at 25,000× *g* for 5 min. The collected supernatant was concentrated to 50 μL under N stream and then stored under −80 °C. For IAA, after derivatization of the supernatant, the organic phase was dried with Na_2_SO_4_. For both GA_3_ and CK, the supernatants (50 μL each) were derivatized and dried with N-methyl-N-(trimethylsilyl)trifluoro acetamide. Identification of IAA, GA_3_, and CK was carried out via comparison of hormone retention time, LRIs, and selected ions with authentic standards. ABA extraction was performed with 100 mM CH_3_OH: CHCl_3_: 2N NH_4_OH (12:5:3, *v*/*v*/*v*), and its level was determined using HPLC [61].

### 2.9. Statistical Analysis Tests

Two-way ANOVA for the split-plot layout was applied to the statistical setup of all obtained observations [62]. Prior to beginning the analysis, data were exposed to a homogeneity test for error variances [63], as well as for normality distribution [64]. A significant difference between each 2 means was estimated at a 5% probability level (*p* ≤ 0.01 and 0.05) using Tukey’s HSD test. The GenStat software, 17th Ed. (VSN International Ltd., Hemel Hempstead, UK) was applied for statistical analysis. Pearson’s correlation coefficients, heat map, and PCA-biplot graphs were executed utilizing R-software (version 4.1.3, https://CRAN.R-project.org, accessed on 29 June 2023).

## 3. Results

This study was carried out to investigate the use of a diluted solution (1.5%) prepared from raw bee honey (DHS) applied as a foliar spray to *Phaseolus vulgaris* plants exposed to three N fertilization rates (100, 125, and 150% of the recommended N dose; RND) as an efficient eco-friendly way to mitigate the detrimental effects of N toxicity on growth, physio-chemical properties, antioxidant defense systems, enzyme gene expression, yield, and pod quality of *P. vulgaris*.

### 3.1. Growth, Yield, and Pod Quality Traits

Figure 1 shows the number of leaves on plant (NLP), area of plant leaves (LAP), dry weight of plant shoot (SDWP), number of green pods (NGPP), weight of pods on plant (GPWP), and green pod yield hectare^−1^ (GPYH) as growth and yield traits of *P. vulgaris*, which were affected by N fertilization rate and DHS application; Figure 1a–f, respectively. The application of 125% RND notably increased NLP (by 24.2%), LAP (by 14.5%), SDWP (by 46.3%), NGPP (by 29.5%), GPWP (by 16.7%), and GPYH (by 9.5%), while the application of 150% RND did not affect these parameters relative to 100% RND (control). Under 100% or 125% RND, foliar spraying with 1.5% DHS markedly enhanced all growth and yield indices of *P. vulgaris* plants compared to 0% DHS (distilled water; control). However, the application of DHS did not influence all growth and yield indices under the application of 150% RND. As the best treatment, “125% RND × foliar spraying with 1.5% DHS” notably elevated NLP, LAP, SDWP, NGPP, GPWP, and GPYH by 6.5, 17.5, 8.2, 8.9, 8.9, and 10.0%, respectively, compared to the controls.

Figure 2 shows the quality traits of *P. vulgaris* pods such as protein content (Figure 2a), carbohydrate content (Figure 2b), fiber content (Figure 2c), N content (Figure 2d), P content (Figure 2e), K content (Figure 2f), NO_3_^−^ content (Figure 2g) and nitrate reductase (NR) activity (Figure 2h) as affected by DHS under different N treatments. Specifically, the application of 125% RND considerably increased pod contents of protein (by 15.3%), carbohydrates (by 16.1%), fibers (by 11.6%), N (by 9.4%), P (by 15.9%), and K (by 15.6%), while using 150% RND did not affect these quality traits compared to 100% RND. Under 100% and 125% RND, foliar application of DHS notably increased protein, carbohydrate, fiber, N, P, and K contents in pods compared to foliar spraying with distilled water (0% DHS). However, the application of DHS did not influence all growth and yield indices under the application of 150% RND. As the best treatment, “125% RND × foliar spraying with 1.5% DHS” markedly increased pod protein, carbohydrate, fiber, N, P, and K contents by 8.0, 9.2, 9.1, 11.3, 8.8, and 6.6%, respectively, compared to the controls. In addition, pod NO_3_^−^ content and NR activity progressively increased with the increase in N fertilizer. Applying 125% and 150% RND increased pod NO_3_^−^ content and NR activity by 16.8 and 36.8%, and 47.9 and 48.4%, respectively, relative to 100% RND. Under 100%, 125%, and 150% RND, foliar application of DHS notably decreased pod NO_3_^−^ content and NR activity by 50.4 and 57.5%, 45.9 and 51.9%, and 26.6 and 33.1%, respectively. This means that whenever the rate of N fertilization exceeds 100%, the ability of the DHS to reduce pod NO_3_^−^ content and NR activity decreases.

### 3.2. Photosynthetic Machinery Efficiency

Figure 3 represents leaf photosynthetic efficiency traits, including total chlorophyll (Chl, Figure 3a) and carotenoid (Carts, Figure 3b) contents, Fv/Fm (Figure 3c), and performance index (PI, Figure 3d) of *P. vulgaris* as affected by DHS under elevated nitrogenous fertilization. The application of 125% RND notably improved Chl content (by 9.7%), Carts content (by 13.2%), Fv/Fm (by 6.3%), and PI (by 11.2%), while the application of 150% RND did not affect these parameters compared to 100% RND. Under 100% and 125% RND, foliar application of 1.5% DHS markedly increased Chl and Carts contents, Fv/Fm, and PI in *P. vulgaris* plants compared to 0% DHS. However, the application of DHS did not influence the tested photosynthesis indices under the 150% RND application. As the best treatment, “125% RND × foliar spraying with 1.5% DHS” notably elevated Chl and Carts contents, Fv/Fm, and PI by 11.9, 11.6, 6.0, and 11.8%, respectively, compared to the controls.

### 3.3. Leaf Mineral Contents and Integrity

Figure 4 represents the mineral nutrient (N, P, K, Fe, Mn, and Zn) contents of *P. vulgaris* leaves (Figure 4a–f, respectively) as affected by excessive nitrogenous fertilizer and foliar application of DHS. Specifically, the application of 125% RND considerably increased the contents of P (by 18.9%), K (by 23.8%), Fe (by 13.2%), Mn (by 15.7%), and Zn (by 17.5%), while the application of 150% RND did not affect these nutrients compared to 100% RND. Under 100% and 125% RND, foliar application of 1.5% DHS notably increased the nutrient contents in *P. vulgaris* plants compared to 0% DHS. However, the application of DHS did not affect all of these nutrient contents under the application of 150% RND. As the best treatment, “125% RND × foliar spraying with 1.5% DHS” markedly increased P, K, Fe, Mn, and Zn contents by 10.7, 10.8, 6.8, 10.2, and 7.2%, respectively, compared to the controls. In addition, leaf N content progressively increased with the increase in N fertilizer. Applying 125% and 150% RND increased N content by 31.4 and 55.6%, respectively, relative to 100% RND. Under 100% and 125% RND, foliar application of DHS notably increased N content by 17.2 and 9.9%, respectively. However, the DHS application did not affect N content under 150% RND.

The stability of *P. vulgaris* leaf tissue was evaluated by determining the relative water content (RWC, Figure 5a) and membrane stability index (MSI, Figure 5b). The application of 125% RND markedly increased RWC (by 4.9%) and MSI (by 5.7%), while applying 150% RND did not affect these parameters relative to 100% RND. Under 100% and 125% RND, foliar spraying with 1.5% DHS significantly improved leaf RWC and MSI compared to 0% DHS. However, the application of DHS did not affect RWC and MSI under the application of 150% RND. As the best treatment, “125% RND × foliar spraying with 1.5% DHS” notably increased RWC and MSI, respectively, by 4.9 and 7.7% relative to the controls.

### 3.4. Oxidative Stress Damage and Biomarkers

Figure 6 and Figure 7 display the oxidative stress damage evaluated as electrolyte leakage (EL, Figure 6a) and malondialdehyde (MDA, Figure 6b), stress indicators (hydrogen peroxide (H_2_O_2_, Figure 6c) and superoxide (O_2_^•−^, Figure 6d)), as well as NO_3_^−^ content (Figure 7a) and nitrate reductase (NR) activity (Figure 7b). The levels of EL, MDA, H_2_O_2_, and O_2_^•−^, as well as NO_3_^−^ content and NR activity progressively increased with increasing N fertilizer. Applying 125% and 150% RND increased EL by 14.8 and 36.6%, MDA level by 19.2 and 65.4%, H_2_O_2_ level by 17.4 and 45.8%, O_2_^•−^ level by 25.8 and 87.9%, NO_3_^−^ content by 16.2 and 36.4%, and NR activity by 47.9 and 47.8%, respectively, compared to 100% RND. Under the three N levels (100%, 125%, and 150% RND), foliar application of DHS notably decreased EL levels by 25.4, 25.2, and 27.8%, MDA levels by 26.9, 27.4, and 40.7%, H_2_O_2_ levels by 20.0 22.2, and 32.7%, O_2_^•−^ levels by 13.6, 26.5, and 46.0%, NO_3_^−^ content by 53.0, 47.0, and 26.9%, and NR activity by 52.1, 52.3, and 32.5%, respectively. As a result, when the N rate exceeds 100%, the ability of the DHS to reduce leaf NO_3_^−^ content and NR activity decreases, while its ability to reduce the levels of EL, MDA, H_2_O_2_, and O_2_^•−^ increases.

### 3.5. Osmoregulatory Compounds (ORCs), Non-Enzymatic and Enzymatic Antioxidants

Figure 8 and Figure 9 reveal the ORCs and antioxidant contents (soluble sugars (sugar), proline, ascorbate (AsA), glutathione (GSH), and α-tocopherol (αToc); Figure 8a–d, respectively), and total soluble proteins (protein, Figure 9e), as well as SOD, CAT, APX, and GR activities (Figure 9a–d, respectively) in *P. vulgaris* leaves under N fertilization rates and DHS application. Using 125% RND decreased sugar content by 20.9% and increased protein content by 11.3%, while applying 150% RND did not affect these parameters relative to 100% RND. Under 100% and 125% RND, foliar application of 1.5% DHS markedly improved sugar and protein contents in *P. vulgaris* plants compared to 0% DHS. However, the application of DHS did not affect these parameters under the application of 150% RND. Under 125% RND, foliar application of DHS notably increased sugar and protein contents by 54.9 and 15.3%, respectively, compared to the controls.

In addition, the contents and activities of proline, AsA, GSH, αToc, SOD, CAT, APX, and GR progressively raised with increasing N fertilizer. Applying 125% and 150% RND increased the contents of proline by 33.3 and 100.0%, AsA by 12.8 and 53.7%, GSH by 19.2 and 87.2%, and αToc by 10.7 and 34.6%, respectively. They also increased the activities of SOD by 16.5 and 53.1%, CAT by 23.2 and 87.5%, APX by 17.9 and 70.5%, and GR by 27.6 and 109.9%, respectively, compared to 100% RND. Under applying 100% and 125% RND, foliar spraying with DHS notably increased the contents of proline, AsA, GSH, and αToC by 33.3 and 18.8%, 13.4 and 21.1%, 20.5 and 34.4%, and 10.3 and 12.4%, respectively, as well as increased the activities of SOD, CAT, APX, and GR by 16.5 and 16.8%, 24.4 and 26.6%, 16.5 and 20.5%, and 30.3 and 29.9%, respectively. However, the application of DHS did not affect these parameters under the application of 150% RND.

### 3.6. Gene Transcriptional Level of Antioxidant Enzymes

Figure 10 represents the expression levels of *PvSOD* (Figure 10a), *PvCAT* (Figure 10b), *PvAPX* (Figure 10c), and *PvGR* (Figure 10d) in *P. vulgaris* leaves under N fertilization rate and DHS application. The relative expressions of these genes were concordant and followed the same trend of SOD, CAT, APX, and GR activities. The expression levels of *PvSOD*, *PvCAT*, *PvAPX*, and *PvGR* in *P. vulgaris* leaves progressively increased with the increase in N fertilizer. Applying 125% and 150% RND increased the relative expression of *PvSOD* by 26.0 and 78.0%, *PvCAT* by 22.0 and 74.0%, *PvAPX* by 24.0 and 82.0%, and *PvGR* by 24.0 and 68.0%, respectively, compared to 100% RND. Under 100% and 125% RND, the application of DHS notably increased the relative expression of *PvSOD* by 28.0 and 22.2%, *PvCAT* by 22.0 and 54.6%, *PvAPX* by 26.0 and 24.2%, and *PvGR* by 28.0 and 16.1%, respectively. However, the application of DHS did not affect these parameters under the application of 150% RND.

### 3.7. Phytohormone Contents

Figure 11 displays the hormonal contents (IAA (Figure 11a), GA_3_ (Figure 11b), CK (Figure 11c), and ABA (Figure 11d)) of *P. vulgaris* plants grown under excessive nitrogenous fertilization and DHS application. The application of 125% RND markedly increased the contents of IAA (by 15.2%), GA_3_ (by 18.7%), and CK (by 28.4%), while applying 150% RND did not affect these hormones relative to 100% RND. Under 100% and 125% RND, foliar application of 1.5% DHS markedly improved the contents of these hormones in *P. vulgaris* plants compared to 0% DHS. However, the application of DHS did not affect IAA, GA_3_, and CK contents under the application of 150% RND. As the best treatment, “125% RND × foliar spraying with 1.5% DHS” notably increased IAA, GA_3_, and CK contents by 32.3, 22.2, and 23.7%, respectively, compared to the controls. In addition, ABA content progressively increased with the increase in N fertilizer. Applying 125% and 150% RND increased ABA content by 64.6 and 158.3% compared to 100% RND. Under 100%, 125%, and 150% RND, foliar application of DHS notably decreased ABA content by 32.5, 49.4, and 61.4%, respectively. As a result, whenever the rate of N fertilization exceeds 100%, the ability of the DHS to reduce ABA content increases.

### 3.8. Correlation Analysis between the Studied Traits

Figure 12 represents the associations among different variables of *P. vulgaris* plants influenced by excessive N addition and foliar supplementation of DHS applying Person’s correlation analysis. The current outcomes showed that green pod quality attributes including the contents of protein, carbohydrates, K, P, N, and fibers, green pod yield, green pod weight, number of green pods, and growth parameters including the number of leaves and leaf area were correlated positively (adjusted *p*-value ≤ 0.05) with RWC, chlorophyll, Fv/Fm, PhAc, carotenoids, PI, MSI, and contents of Zn, K, Fe, P, CK, Mn, IAA, GA_3_, and soluble protein in leaves. Meanwhile, O_2_^•−^, ABA, H_2_O_2_, MDA, EL, pod NR activity, leaf NR activity, and pod and leaf NO_3_^−^ contents were negatively correlated with the above-mentioned variables. Additionally, APX, GR, CAT, and SOD activities, gene expression, and contents of αToC, GSH, AsA, and proline revealed a significant (adjusted *p*-value ≤ 0.05) positive correlation with each other.

Figure 13 represents a heat map with hierarchical analysis revealing the interactive connection between different variables of *P. vulgaris* plants influenced by excessive N addition and foliar supplementation of DHS. Using the hierarchical cluster analysis, treatments were separated into three primary sets (X-axis). The first main group (I) included 100%RND_0.0%DHS, 100%RND_1.5%DHS, and 125%RND_0.0%DHS; the second primary group (II) contained 150%RND_0.0%DHS and 150%RND_1.5%DHS; the third main group (III) represented 125%RND_1.5%DHS. Moreover, T3; 100%RND_1.5%DHS and T2; 125%RND_0.0%DHS were clustered together at the sub-group level of clustering in the first group (I). These findings indicated that foliar spraying with DHS enhanced the growth, physio-biochemical attributes, and green pod quality traits of *P. vulgaris* under excessive nitrogenous fertilizer. In addition, the studied variables (Y-axis) were separated into three main groups (1, 2, and 3) based on the hierarchical cluster analysis. These results indicated that different treatments have different influence patterns on the assessed parameters or variables.

Figure 14 represents a principal component analysis (PCA) biplot to explore the high variations because of the impact of excessive nitrogenous fertilizer and foliar supplementation of DHS on the studied variables. Dim 1 and Dim 2 (PCA-diminution 1 and 2, respectively) explored 62.7% and 25.1% variability of data, respectively. The large variability between T1, 100%RND_0.0%BHS and T2, 100%RND_1.5%BHS and also between T3, 125%RND_0.0%BHS and T4, 125%RND_1.5%BHS indicated the role of DHS in promoting growth, physio-biochemistry, and green pod quality traits under excessive nitrogenous fertilizer. Foliar spraying with DHS improved the contents of soluble protein, IAA, chlorophyll, pod fiber, GA_3_, pod N, CK, leaf Mn, carotenoids, leaf Fe, leaf K, leaf P, pod P, pod protein, leaf Zn, pod protein, and pod carbohydrates. The variables MSI, RWC, Fv/Fm, green pods weight, and green pods yield that clustered together were also promoted by DHS. Additionally, the PCA biplot indicated that the above-mentioned parameters were associated with T4, 125%RND_1.5%BHS that showed a positive influence on the yield and growth of *P. vulgaris* plants under excessive nitrogenous fertilizer, while parameters of O_2_^•−^, ABA, H_2_O_2_, MDA, EL, pod NR activity, leaf NR activity, pod NO_3_^−^ content, and leaf NO_3_^−^ content were associated with T5, 150%RND_0.0%BHS. Therefore, foliar spraying with DHS has a distinct role in improving the green pod quality and biomass production and overcoming the excess N stress in *P. vulgaris* plants.

## 4. Discussion

Because N is required for good growth and development, it is an essential protein component used to build the substances of various plant cells and tissues [11]. However, excessive N application could be toxic for plants. The toxicity is thought to be caused by a trigger of endless generation of ROS and an elevation in NO_3_^−^ content in plant tissues (Figure 6 and Figure 7). Plants can counterbalance ROS levels under favorable conditions as the excess amount gets scavenged by different mechanisms, including non-enzymatic and enzymatic antioxidant systems [9,13].

The use of bee honey (as a diluted honey solution; DHS) in agriculture has recently received awareness because it has potential benefits for plant development and growth, owing to its many bioactive compounds and plant growth regulators as effective mechanisms (Table 1). These compounds can protect plants from ROS-stimulated oxidative stress, stimulate plant growth, and enhance nutrient absorption and utilization under abiotic stresses [17,18,25,26,65]. These reports showed that foliar spraying with DHS at a level of 0.5–1.5% for drought-stressed *Phaseolus vulgaris* plants [17], saline-calcareous-stressed *Atriplex nummularia* plants [18], drought-stressed *Vicia faba* plants [26], and salt-stressed *Allium cepa* and *Capsicum annuum* plants [25,65] effectively attenuates the deleterious impacts of stress. The reports attributed these positive findings to marked improvements in antioxidant defense systems, including non-enzymatic antioxidants, antioxidant enzymes, and enzyme gene expressions in association with osmoregulatory compounds (ORCs), all of which stimulate the minimization of ROS-catalyzed oxidative stress. This leads to remarkable improvements in plant leaf integrity, photosynthetic efficiency, various nutrient contents, and hormonal homeostasis, and thus noticeable improvements in plant growth, productivity, and production quality. However, the potential of DHS to mitigate the excessive nitrogenous stress in *Phaseolus vulgaris* L. plants has not been studied yet. Therefore, in this study, DHS was used to treat *P. vulgaris* L. plants exposed to excessive N application to investigate its benefits in mitigating the harmful influences of excess N on plants (Figure 1, Figure 2, Figure 3, Figure 4, Figure 5, Figure 6, Figure 7, Figure 8, Figure 9, Figure 10 and Figure 11).

This study indicates that excessive N application negatively affected *P. vulgaris* growth and yield (Figure 1), as well as the quality of green pods (Figure 2) due to increased levels of EL and MDA (Figure 6) due to elevated levels of NO_3_^−^, H_2_O_2_, and O_2_^•−^ (oxidative damage; Figure 6 and Figure 7). However, foliar spraying with 1.5% DHS restored growth and pod quality traits, and the best quality improvements were obtained under 125% of the recommended N dose (125% RND) due to minimal levels of oxidative stress markers. These findings indicate that DHS plays a major role in plant growth, productivity, and yield quality under excessive N utilization by providing plants with various growth biostimulators (BSs: soluble sugars, proline, necessary minerals, different antioxidants, and B-group vitamins; Table 1) as useful mechanisms against excessive N stress.

These DHS components can help plants to withstand and recover from environmental stressors [17,24,25,65]. Furthermore, these findings align with prior studies indicating that using soluble sugars as BSs has a central regulatory function in steering plant growth, and can significantly enhance crop growth and yield while also impacting the capacity of plants to cope with environmental stress factors [25,66,67,68,69,70,71,72,73,74]. The same central roles are well-proven for antioxidants, including proline, and the nutrients have been demonstrated as effective mechanisms in elevating the ability of plants to withstand environmental stressors [25,26,65]. Encoring growth, yield, and pod quality of excessive N-stressed *P. vulgaris* plants by foliar spraying with DHS indicated that this multi-bio-stimulator (mBS; DHS) has multiple mechanisms (e.g., osmoprotectants, different antioxidants, including ascorbate and B-group vitamins, and various nutrients, including iodine and selenium). All of these BSs support plants to encore their development and growth because of the mitigation of excessive N stress [16,17,24].

Stress conditions stimulated a decrease in chlorophyll content and chlorophyll fluorescence, resulting in decreased photosynthesis [75,76,77,78,79,80,81]. Leaf contents of chlorophyll and carotenoids and the efficiency of photosynthesis (PI and Fv/Fm) of *P. vulgaris* were negatively affected under excessive N stress (Figure 3). This negative outcome may have been obtained as a result of increasing ROS, including H_2_O_2_ and O_2_^•−^ (Figure 6), which coincided with increasing EL and MDA levels (Figure 6) and decreasing leaf RWC and MSI (Figure 5), as well as nutrient contents (Figure 4). The application of N can change the plant’s adaptation range to light and reduce the plant’s ability to adapt to the light environment, so the excessive application of N to plants does not enhance the plant’s light use efficiency [10], which negatively affects photosynthesis efficiency traits (Figure 3). However, the photosynthetic efficiency traits were markedly increased by foliar supplementation of DHS, and this could be attributed to the fact that mBS (DHS) contains bioactive organic compounds (Table 1) as beneficial stress mitigation mechanisms, including Mg. This nutrient (Mg) has the potential to increase chlorophyll and carotenoid contents, and chlorophyll performance/fluorescence, resulting in enhancing photosynthesis and thus the plant’s defenses against stress [25,26,82,83,84,85,86]. In this study, the increase in nutrients (Figure 4) and RWC (Figure 5) seen after the application of DHS helped to create an environment suitable for healthy photosynthesis.

As beneficial stress mitigation mechanisms, nutrients in appropriate concentrations maintain the ion balance between cells to promote chlorophyll biosynthesis in *P. vulgaris* leaves in favor of stimulating photosynthesis. Another stress mitigation mechanism, various soluble sugars (as ORCs) in DHS can raise RWC in favor of healthy plant metabolism. Furthermore, different antioxidants and vitamins present in DHS entering plant leaves after spraying act as powerful beneficial stress-alleviating mechanisms to support antioxidant defense systems in *P. vulgaris* plants [17]. In this study, enhanced photosynthesis indices were coupled with higher proline content (Figure 3 and Figure 8). As reported recently [17], proline is a larger substance that accumulates in stressed plants, which contribute as a powerful and beneficial stress-alleviating mechanism to the efficiency of photosynthesis and the generation of ATP compounds. In this context, plant nourishment with DHS further increased the antioxidant proline content under excessive N stress (Figure 8). In addition, in this study, enhanced ORCs, including proline and soluble sugars (effective stress alleviation mechanisms), resulting with the application of DHS under excessive N stress conditions may corroborate their relationship with increased tolerance to excessive N stress in *P. vulgaris* plants [87].

Figure 4 shows that the application of N at 125% RND markedly increased the contents of all nutrients tested, whereas excessive application of N fertilizer (150% RND) did not affect all nutrients [88] but negatively affected leaf tissue cell integrity (Figure 5). Nonetheless, plant nourishment with DHS stimulated increased nutrient levels, especially under 125% RND. This result could be attributed to the richness of DHS in essential nutrients (Table 1), which represent many stress alleviation mechanisms [89,90]. DHS nourishment caused ionic homeostasis and increased nutrient contents under excessive N stress (Figure 4). This could be due to enhanced expansion of absorbing root surfaces due to the augmented size of the root system, and/or improved accumulation of ORCs (Figure 8) to balance cellular osmotic pressure. Therefore, cell turgor is sustained and nutrient uptake is improved for the benefit of RWC [17]. This could also be due to an improved photosynthesis process resulting in the accumulation/production of osmotically active compounds (ORCs), which together with those absorbed from the DHS enhance water efficiency and the RWC of leaves. Ultimately, this can improve the absorption of mineral nutrients, water translocation, and root efficiency [91,92]. RWC is a physiological measure that indicates the availability of water in metabolizing tissues, while MSI and EL indicate the status of cell membrane integrity [26]. The recovery of the stressed leaf tissues [i.e., maintained cell turgor (RWC) and membrane integrity (MSI) with minimal EL] was mediated by DHS in this research (Figure 5 and Figure 6). Plant nourishment with DHS markedly improved RWC and MSI in excessive N-stressed *P. vulgaris* leaves, and sustained cell development through the accumulation of more ORCs (Figure 8). This indicated the continued activity of metabolic processes as effective mechanisms of tolerance to N stress in *P. vulgaris* plants.

In this study, excessive N application resulted in membrane damage, increased generation of H_2_O_2_, MDA, and O_2_^•−^ (Figure 6), and increased NO_3_^−^ content and NR activity in leaves (Figure 7). Interestingly, DHS minimized this trend as evidenced by decreased H_2_O_2_ MDA, O_2_^•−^, and NO_3_^−^ levels, and NR activity. Based on our findings, DHS exhibited protective properties against membrane damage in the presence of excessive N-induced stress, and similar conclusions have been affirmed using other plants under oxidative stress [25,26,93,94]. In this research, the ORCs, different antioxidant enzymes and antioxidant compounds (Figure 8 and Figure 9) were considerably promoted by DHS application to protect cellular plasma membranes from oxidants, including H_2_O_2_ and O_2_^•−^ by minimizing lipid peroxidation (MDA), and EL by minimizing the levels of these oxidants (Figure 6). These positive findings led to elevated MSI, reduced EL and photo-oxidation, and promoted membrane integrity under oxidative stress [17,26]. In addition, DHS had the ability to markedly reduce NO_3_^−^ content due to the reduction in NR activity (Figure 7) and the maximization of different antioxidant activities (Figure 8 and Figure 9), all in favor of enhancing *P. vulgaris* plant growth, yield, and pod quality under excessive N stress.

As presented in Figure 8 and Figure 9, excessive N stress markedly increased the activity of ORCs and enzymes [SOD (superoxide dismutase), CAT (catalase), APX (ascorbate peroxidase), and GR (glutathione reductase)] and antioxidant compounds [soluble sugars, soluble protein, proline, AsA (ascorbate), GSH (glutathione), αToc (α-tocopherol)], whose activities further increased with DHS application. Ascorbate and GSH have several antioxidative properties, and have active roles in protective defenses against different stresses, including excessive N stress, through the contribution of electrons in several reactions related to different antioxidants. As an effectual eliminator of ROS, AsA directly scavenges O_2_^•−^ and OH^−^ and protects stressed cell membranes [17]. In this study, increased contents of AsA, and GSH under excessive N stress contributed to the DHS-stimulated decrease in levels of ROS (O_2_^•−^ and H_2_O_2_), EL, and MDA. Thus, the equilibrium of AsA and GSH pools ought to be precisely upregulated with sufficient APX activity, which in this study was also improved by DHS (Figure 8 and Figure 9) to elevate the antioxidant capacity of cells and prevent damage from oxidative stress [95]. Increasing AsA and GSH levels due to DHS application is proposed to help complete the “AsA-GSH cycle” to regulate cell H_2_O_2_ amount. In association with DHAR and MDHAR, GR initially provides substrates for APX through the continuous generation of AsA and GSH [17,26]. Under stress, the increase in αToc content (a non-enzymatic lipophilic antioxidant; stress alleviation mechanism) with DHS application can eliminate several ROS, and reduce indicators of oxidative stress (O_2_^•−^ and H_2_O_2_) and levels of EL and MDA to keep plasma membrane integrity [17]. As a specific target of many oxidants, plasma membrane repair is prominently mediated by αToc which minimizes lipid peroxidation by limiting the development of oxidized phospholipids, which may interfere with membrane fusion processes. These antioxidant compounds offer the ability to withstand excessive N stress, which stimulates ROS generation at different stages of development [96]. Plant nourishment with DHS can suppress damage to cell membranes by perfecting the photosynthesis machinery, elevating ROS elimination, improving morphological and physiological indicators, and upregulating antioxidant defense systems, as shown in this study (Figure 1, Figure 2, Figure 3, Figure 4, Figure 5, Figure 6, Figure 7, Figure 8, Figure 9, Figure 10 and Figure 11). Subsequently, DHS (an mBS with multiple stress alleviation mechanisms) can promote the development and productivity of excessive N-stressed *P. vulgaris* plants, and have positive impacts on farming productivity features. Also, SOD has a crucial role in removing ROS, and acts as a primary defense mechanism in the antioxidation systems [82] by converting O_2_^•−^ into H_2_O_2_, which is then rapidly detoxified by CAT, transforming it into O_2_ and H_2_O [97]. The enzymes POD and APX play the same role as CAT [83]. These enzymes increase the ability to remove ROS and free radicals, protecting cells from oxidative stress and minimizing stress damage. This paper showed that both ORCs and non-enzymatic antioxidant contents are affected under conditions of excessive N utilization (Figure 8 and Figure 9). More important, our results showed that foliar supplementation of DHS significantly enhanced the antioxidant defense systems that were able to reduce the oxidative damage triggered by excessive N stress by boosting the activities of enzymes, thereby balancing ROS production and decreasing the damage to PSII. Our findings agree with previous studies on other plants [21,94,98,99].

At the molecular scale, the expression level of antioxidant enzyme genes, including *PvSOD, PvCAT, PvAPX*, and *PvGR* (Figure 10), was correlated with the activity of the corresponding enzymes (Figure 9). Antioxidant enzyme activities increased due to increased expression of these genes under increasing N levels, and DHS application further augmented the SOD, CAT, APX, and GR enzyme activities owing to the further up-regulation of *PvSOD, PvCAT, PvAPX*, and *PvGR* genes. These findings suggest that changes in gene transcription can serve as an indicator of enzyme activity in response to DHS, which is able to effectively counterbalance ROS levels, eliminating the excess ROS amounts and preventing their buildup to harmful concentrations. These outcomes are consistent with [100,101,102,103,104,105,106,107].

Another point to be considered, Figure 11 shows that the application of 125% RND resulted in the highest increases in levels of the phytohormones tested [IAA (indole acetic acid), GA_3_ (gibberellic acid), and CK (cytokinins)]. Excessive application of N fertilizer (150% RND) did not affect phytohormone contents, but significantly increased the abscisic acid (ABA) content. Nonetheless, the use of DHS resulted in a further increase in IAA, GA_3_, and CK levels, while ABA levels decreased further, especially under 125% RND. These findings indicate that the improved hormone levels efficiently regulated *P. vulgaris* plant growth under excessive N stress (Figure 1). These phytohormones, along with antioxidants, play crucial roles as powerful and useful stress reduction mechanisms by helping in mitigating and repairing damage caused by ROS during stress, resulting in the positive regulation of plant performance under excessive N stress. These findings were confirmed in other studies under different stresses [85,108,109].

As reported in this study, excessive N-stressed *P. vulgaris* plants need an exogenous adjuvant with effective growth BSs that can induce the plant’s physiology- and biochemistry-related processes, reflecting positive regulation of plant growth, production quality, and plant tolerance to excessive N stress. Table 1 reveals the richness of DHS (mBS) in ORCs, and various nutrients including selenium and iodine, B-group vitamins, and vitamin C. Also, DHS has strong DPPH-radical scavenging activity (89.4%). Therefore, DHS has a high antioxidant capacity to prevent, or at least minimize, lipid peroxidation [17]. A previous report [25] indicates that DHS has a minimum H_2_O_2_ concentration. In the present study, this mechanism may have contributed to enhanced tolerance to excessive N stress in *P. vulgaris* plants. Therefore, DHS possesses multiple crucial stress-mitigating mechanism, and can counter several reactions related to cell metabolism to overcome abiotic stress [17,18,24,25,65], including excessive N stress. In addition, DHS can be used as a nutrient solution because of its nutrients, amino acids, sugars, and vitamins, all of which support plants that thrive under stressful conditions. Therefore, DHS is an inexpensive and easy-to-prepare natural tool that can replace expensive synthetic tools. Previously, all bioactive components of DHS have been applied individually as foliar sprayings with great success, and they helped plants of various crops to perform well due to increased plant stress tolerance. For example, soluble sugars [73], amino acids [110,111], proline [87,110,112], different nutrients [113,114,115], ascorbate [116], B-group vitamins [117], selenium [118], and iodine [118,119] are applied individually for different stressed plants. In this study, all of these vital organic substances, and others, are combined in just one solution (DHS), conferring DHS with multiple stress-attenuating mechanisms to effectively support *P. vulgaris* plants to adapt to and withstand excessive N stress conditions. This occurred through improvement in signaling pathways and introduction of new mechanisms (e.g., osmoregulation, organic acids, antioxidant systems, nutrition, etc.) with the application of DHS. These multiple stress attenuation mechanisms found in this study have proven DHS as a multi-growth bio-stimulator for *P. vulgaris* plants grown under excessive N stress with significant success.

## 5. Conclusions

Bee honey is rich in antioxidants, osmotically active substances, vitamins, and various nutrients, including selenium and iodine, all of which make DHS an effective eco-friendly policy that can be used to mitigate the unwanted influences of abiotic stress, including excessive N stress, and help to sustain crops under stress. Therefore, nourishing plant leaves with diluted bee honey solution (DHS), reduced stress, mitigated the toxicity stimulated by excessive N utilization, and promoted growth, productivity, production quality, photosynthetic efficiency, nutritional and hormonal balance, leaf integrity, osmotically active substances, antioxidant activities, and enzyme gene expression, thus enhancing antioxidant defense systems to effectively eliminate ROS for plant survival under stress. Although DHS has been previously applied to certain stressed crops (e.g., *Phaseolus vulgaris* and *Vicia faba* under drought stress, *Atriplex nummularia* under saline-calcareous stress, and *Allium cepa* and *Capsicum annuum* under salt stress), more research using other crops is still needed to explore the accurate stress-relieving mechanisms of the DHS ingredients. Such research will help in encouraging agricultural producers to use DHS to stimulate the activity of the plant’s antioxidant defense systems to enhance plant stress tolerance, thereby promoting higher plant productivity. However, challenges in the application of DHS may arise in the future. Although no pathogen problems have been detected in utilizing DHS to nourish stressed *Phaseolus vulgaris* plants, further research needs to be conducted to explore this in detail.

## Figures and Tables

**Figure 1 plants-12-03435-f001:**
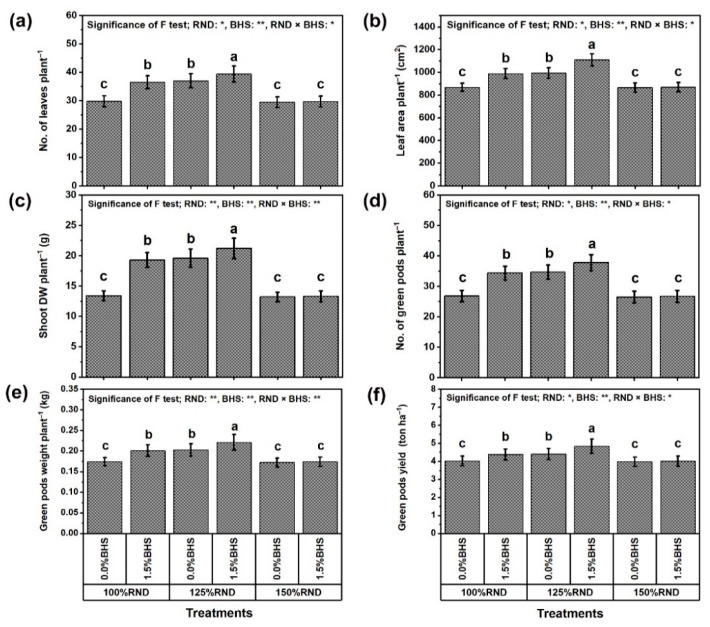
Foliar supplementation influences of BHS (bee honey solution) on growth [(**a**) no. of leaves, (**b**) leaf area, and (**c**) shoot DW] and green pod yield components [(**d**) no. of green pods, (**e**) green pods weight, and (**f**) green pods yield] of *P. vulgaris* plants grown under excessive nitrogenous fertilizer (ammonium nitrate, NH_4_NO_3_). RND, the recommended N fertilizer dose; DW, dry weight. ^(^**^)^ or ^(^*^)^ signalizes differences at *p* ≤ 0.01 or 0.05 level of probability, respectively. In each plot, based on the LSD test (*p* ≤ 0.05), columns (±SE bar) with different letters are significantly different.

**Figure 2 plants-12-03435-f002:**
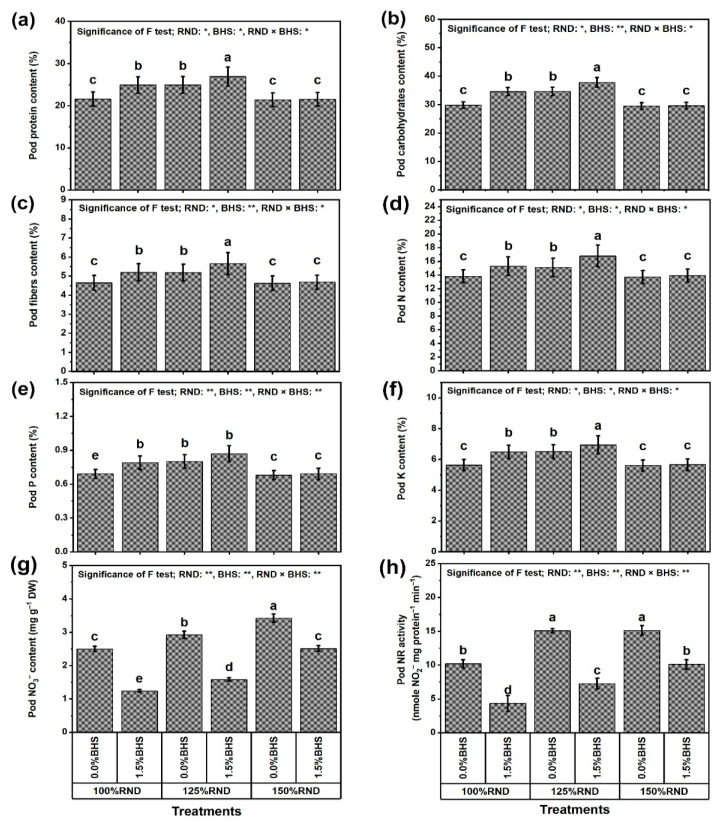
Foliar supplementation influences of BHS (bee honey solution) on pod quality traits [contents of (**a**) protein, (**b**) carbohydrate, (**c**) fibers, (**d**) N, (**e**) P, (**f**) K, (**g**) NO_3_^−^, and (**h**) NR activity] of *P. vulgaris* plants grown under excessive nitrogenous fertilizer (ammonium nitrate; NH_4_NO_3_). RND, the recommended N fertilizer dose; NO_3_^−^, nitrate; NO_2_^−^, nitrite; and NR, nitrate reductase. ^(^**^)^ or ^(^*^)^ signalizes differences at *p* ≤ 0.01 or 0.05 level of probability, respectively. In each plot, based on the LSD test (*p* ≤ 0.05), columns (±SE bar) with different letters are significantly different.

**Figure 3 plants-12-03435-f003:**
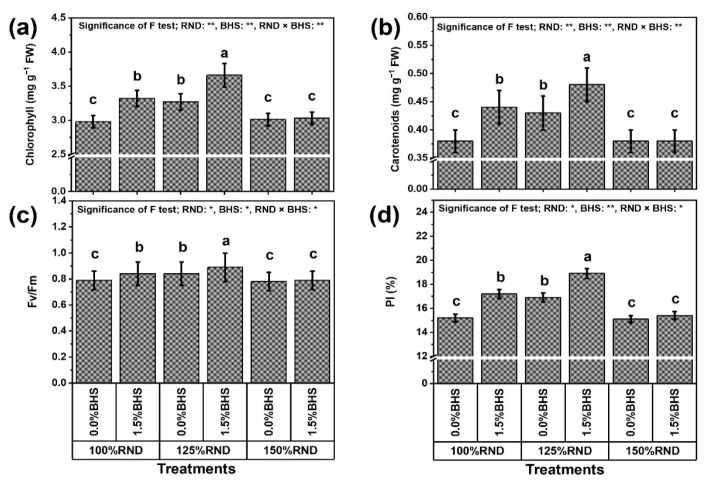
Foliar supplementation influences of BHS (bee honey solution) on photosynthesis indices [(**a**) chlorophyll content, (**b**) carotenoids content, (**c**) Fv/Fm, and (**d**) PI] of *P. vulgaris* plants grown under excessive nitrogenous fertilizer (ammonium nitrate, NH_4_NO_3_). RND, the recommended N fertilizer dose; Fv/Fm, photosystem II quantum efficiency; and PI, performance index. ^(^**^)^ or ^(^*^)^ signalizes differences at *p* ≤ 0.01 or 0.05 level of probability, respectively. In each plot, based on the LSD test (*p* ≤ 0.05), columns (±SE bar) with different letters are significantly different.

**Figure 4 plants-12-03435-f004:**
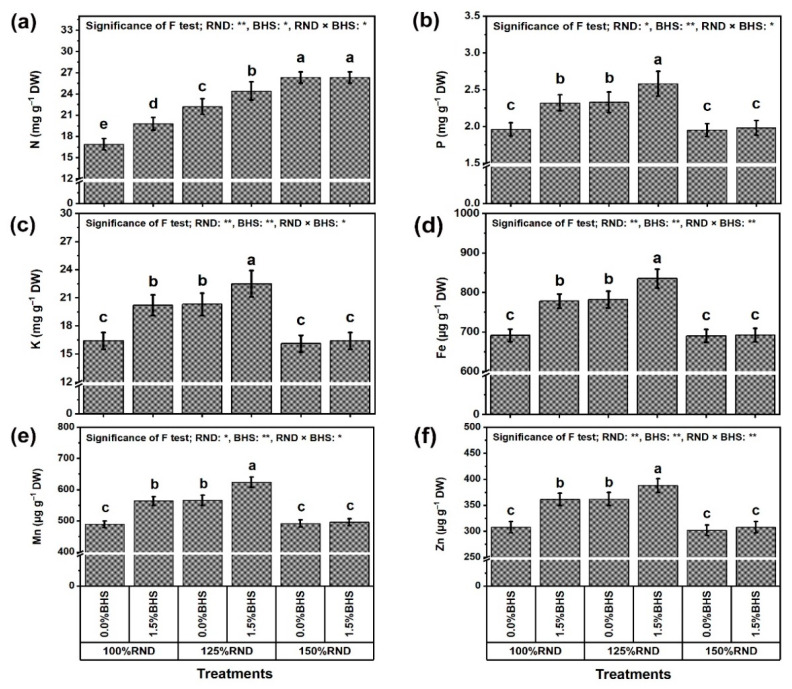
Foliar supplementation influences of BHS (bee honey solution) on leaf nutrient contents [(**a**) N, (**b**) P, (**c**) K, (**d**) Fe, (**e**) Mn, and (**f**) Zn] of *P. vulgaris* plants grown under excessive nitrogenous fertilizer (ammonium nitrate, NH_4_NO_3_). RND, the recommended N fertilizer dose. ^(^**^)^ or ^(^*^)^ signalizes differences at *p* ≤ 0.01 or 0.05 level of probability, respectively. In each plot, based on the LSD test (*p* ≤ 0.05), columns (±SE bar) with different letters are significantly different.

**Figure 5 plants-12-03435-f005:**
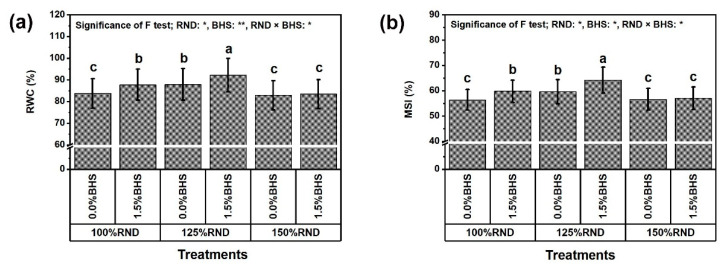
Foliar supplementation influences of BHS (bee honey solution) on the leafy integrity [(**a**) relative water content, RWC; and (**b**) membrane stability index, MSI] of *P. vulgaris* plants grown under excessive nitrogenous fertilizer (ammonium nitrate, NH_4_NO_3_). RND, the recommended N fertilizer dose. ^(^**^)^ or ^(^*^)^ signalizes differences at *p* ≤ 0.01 or 0.05 level of probability, respectively. In each plot, based on the LSD test (*p* ≤ 0.05), columns (±SE bar) with different letters are significantly different.

**Figure 6 plants-12-03435-f006:**
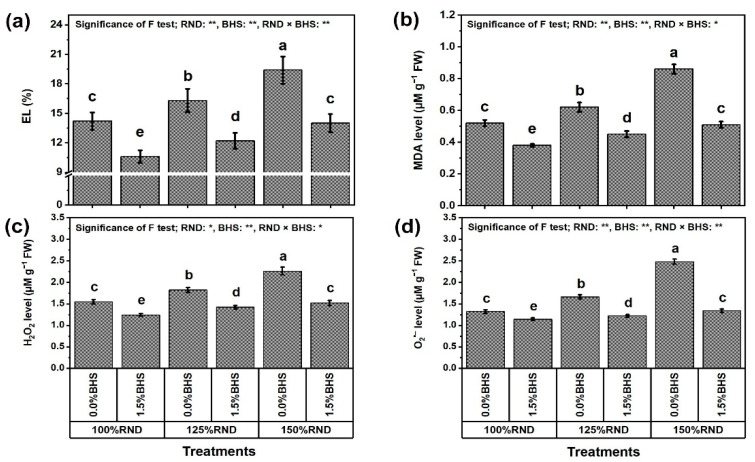
Foliar supplementation influences of BHS (bee honey solution) on oxidative damage [(**a**) electrolyte leakage, EL; and (**b**) malondialdehyde, MDA] and leaf oxidant levels [(**c**) hydrogen peroxide, H_2_O_2,_ and (**d**) superoxide, O_2_^•−^] in *P. vulgaris* plants grown under excessive nitrogenous fertilizer (ammonium nitrate, NH_4_NO_3_). RND, the recommended N fertilizer dose. ^(^**^)^ or ^(^*^)^ signalizes differences at *p* ≤ 0.01 or 0.05 level of probability, respectively. In each plot, based on the LSD test (*p* ≤ 0.05), columns (±SE bar) with different letters are significantly different.

**Figure 7 plants-12-03435-f007:**
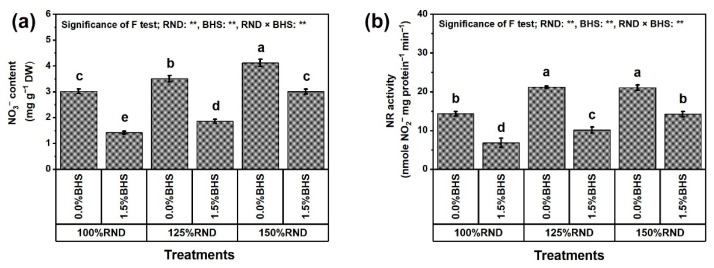
Foliar supplementation influences of BHS (bee honey solution) on (**a**) nitrate, NO_3_^−^ content and (**b**) nitrate reductase, NR activity of *Phaseolus vulgaris* plants grown under excessive nitrogenous fertilizer (ammonium nitrate, NH_4_NO_3_). RND, the recommended N fertilizer dose; NO_3_^−^, nitrate; and NR, nitrate reductase. ^(^**^)^ or ^(^*^)^ signalizes differences at *p* ≤ 0.01 or 0.05 level of probability, respectively. In each plot, based on the LSD test (*p* ≤ 0.05), columns (±SE bar) with different letters are significantly different.

**Figure 8 plants-12-03435-f008:**
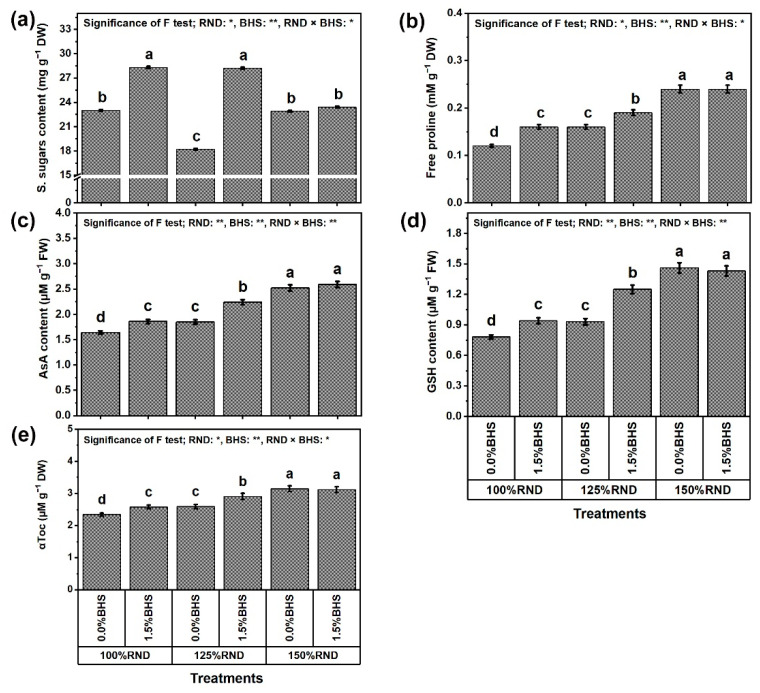
Foliar supplementation influences of BHS (bee honey solution) on osmoprotectant and antioxidant contents [(**a**) soluble sugars, S. sugars; (**b**) free proline; (**c**) ascorbate, AsA; (**d**) glutathione, GSH; and (**e**) α-Tocopherol, αToc] of *P. vulgaris* plants grown under excessive nitrogenous fertilizer (ammonium nitrate, NH_4_NO_3_). RND, the recommended N fertilizer dose. ^(^**^)^ or ^(^*^)^ signalizes differences at *p* ≤ 0.01 or 0.05 level of probability, respectively. In each plot, based on the LSD test (*p* ≤ 0.05), columns (±SE bar) with different letters are significantly different.

**Figure 9 plants-12-03435-f009:**
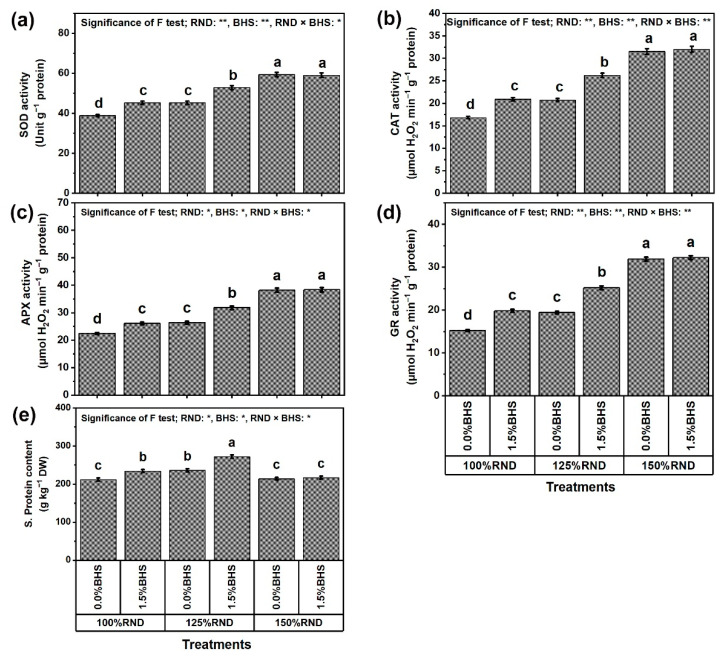
Foliar supplementation influences of BHS (bee honey solution) on the enzyme activity [(**a**) SOD (superoxide dismutase), (**b**) CAT (catalase), (**c**) APX (ascorbate peroxidase), and (**d**) GR (glutathione reductase)] and (**e**) soluble protein content of *P. vulgaris* plants grown under excessive nitrogenous fertilizer (ammonium nitrate, NH_4_NO_3_). RND, the recommended N fertilizer dose. ^(^**^)^ or ^(^*^)^ signalizes differences at *p* ≤ 0.01 or 0.05 level of probability, respectively. In each plot, based on the LSD test (*p* ≤ 0.05), columns (±SE bar) with different letters are significantly different.

**Figure 10 plants-12-03435-f010:**
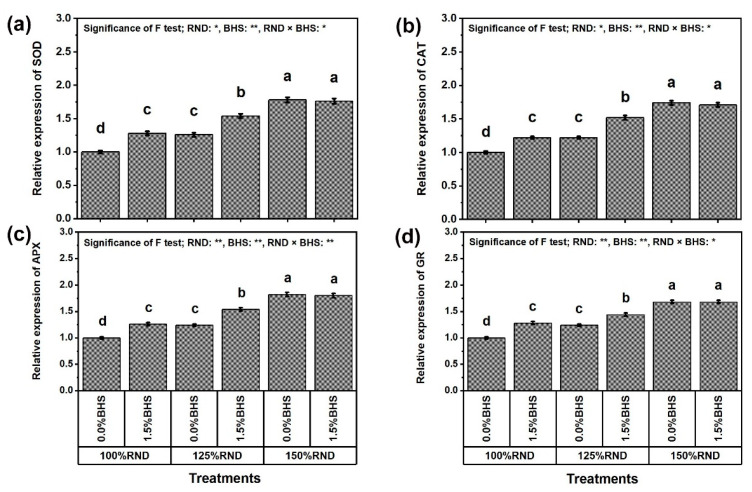
Foliar supplementation influences of BHS (bee honey solution) on the enzyme transcript levels [(**a**) *PvSOD*, (**b**) *PvCAT*, (**c**) *PvAPX*, and (**d**) *PvGR*] encoding genes of *Phaseolus vulgaris* plants grown under excessive nitrogenous fertilizer (ammonium nitrate, NH_4_NO_3_). RND, the recommended N fertilizer dose. ^(^**^)^ or ^(^*^)^ signalizes differences at *p* ≤ 0.01 or 0.05 level of probability, respectively. In each plot, based on the LSD test (*p* ≤ 0.05), columns (±SE bar) with different letters are significantly different.

**Figure 11 plants-12-03435-f011:**
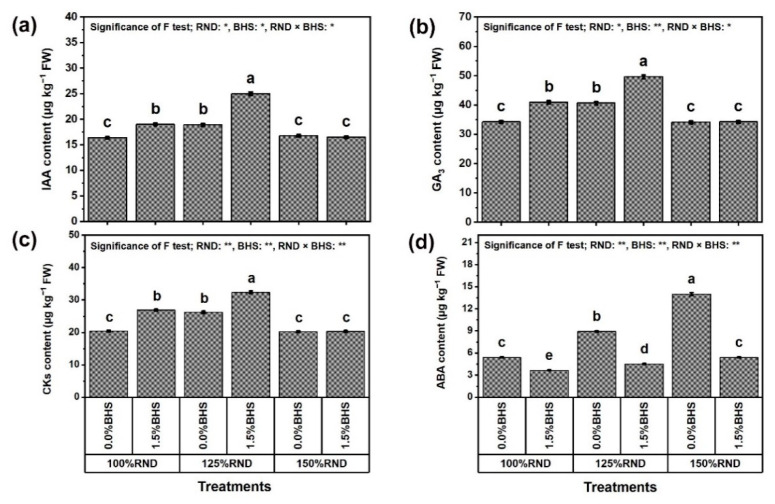
Foliar supplementation influences of BHS (bee honey solution) on hormonal levels [(**a**) IAA (indole-3-acetic acid), (**b**) GA_3_ (gibberellic acid), (**c**) CK (cytokinins), and (**d**) ABA (abscisic acid)] of *P. vulgaris* plants grown under excessive nitrogenous fertilizer (ammonium nitrate, NH_4_NO_3_). RND, the recommended N fertilizer dose. ^(^**^)^ or ^(^*^)^ signalizes differences at *p* ≤ 0.01 or 0.05 level of probability, respectively. In each plot, based on the LSD test (*p* ≤ 0.05), columns (±SE bar) with different letters are significantly different.

**Figure 12 plants-12-03435-f012:**
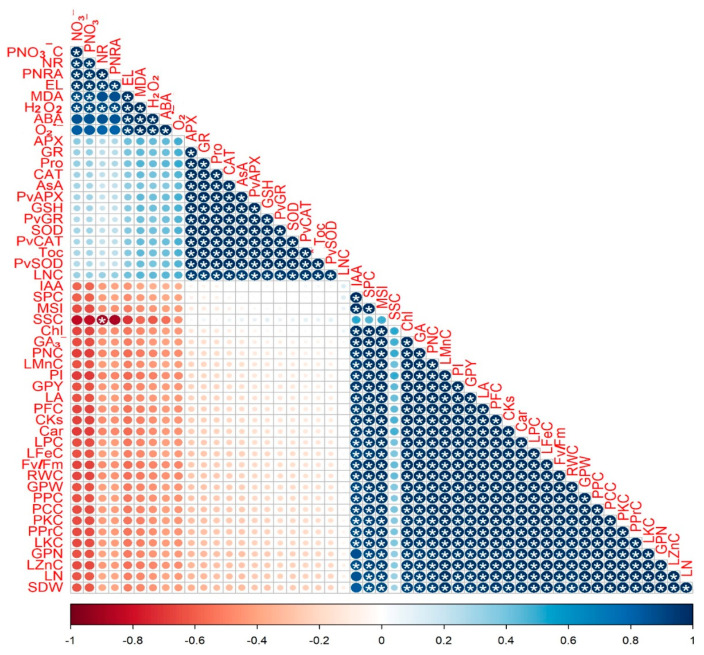
Pearson’s correlation analysis graph for the examined indices. Variations are represented by colors. ^(^*^)^ indicates the significance at *p*-adjusted value (“Bonferroni” method) ≤ 0.05. The parameters number of leaves, leaves area, shoot dry weight, number of pods, pod weight, pod yield, pod protein, carbohydrate, fiber, nitrogen, phosphorous, potassium, and nitrate contents, pod nitrate reductase activity, leaf nitrate, chlorophyll, carotenoid, nitrogen, phosphorus, potassium, iron, manganese, zinc, malondialdehyde, hydrogen peroxide, superoxide, soluble sugar, soluble protein, ascorbate, glutathione, α-tocopherol, indole-3-acetic acid, gibberellic acid, cytokinins, proline, and abscisic acid contents, photosystem II quantum efficiency, performance index, relative water content, membrane stability index, electrolyte leakage, superoxide dismutase, catalase, ascorbate peroxidase, and glutathione reductase activities, and *GR*, *SOD*, *CAT*, and *APX* relative expressions are abbreviated as LN, LA, SDW, GPN, GPW, GPY, PPrc, PCC, PFC, PNC, PPC, PKC, PNO_3_^−^C, PNRA, NO_3_^−^, Chl, Car, LNC, LPC, LKC, LFeC, LMnC, LZnC, MDA, H_2_O_2_, O_2_^•−^, SSC, SPC, AsA, GSH, αToC, IAA, GA_3_, CK, Pro, ABA, Fv/Fm, PI, RWC, MSI, EL, *SOD*, *CAT*, *APX*, *GR*, *PvGR*, *PvSOD*, *PvCAT*, and *PvAPX*, respectively.

**Figure 13 plants-12-03435-f013:**
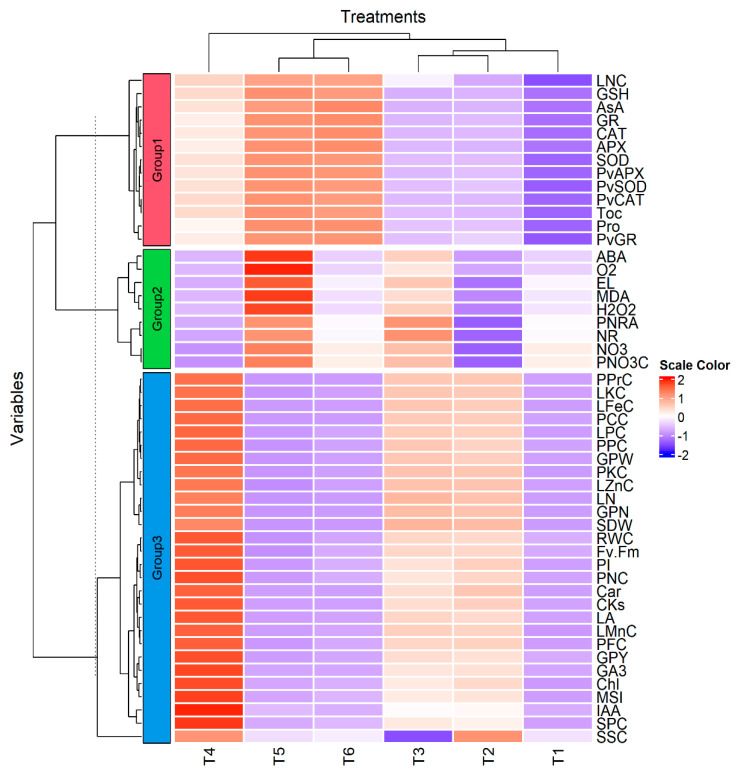
A hierarchical clustering analysis of different examined treatments and parameters using a heat map graph. The variations in the obtained are represented by colors. T1, 100%RND_0.0%BHS; T2, 100%RND_1.5%BHS; T3, 125%RND_0.0%BHS; T4, 125%RND_1.5%BHS; T5; 150%RND_0.0%BHS; and T6, 150%RND_1.5%BHS. The parameters number of leaves, leaves area, shoot dry weight, number of pods, pod weight, pod yield, pod protein, carbohydrate, fiber, nitrogen, phosphorous, potassium, and nitrate contents, pod nitrate reductase activity, leaf nitrate, chlorophyll, carotenoid, nitrogen, phosphorus, potassium, iron, manganese, zinc, malondialdehyde, hydrogen peroxide, superoxide, soluble sugar, soluble protein, ascorbate, glutathione, α-tocopherol, indole-3-acetic acid, gibberellic acid, cytokinins, proline, and abscisic acid contents, photosystem II quantum efficiency, performance index, relative water content, membrane stability index, electrolyte leakage, superoxide dismutase, catalase, ascorbate peroxidase, and glutathione reductase activities, and *GR*, *SOD*, *CAT*, and *APX* relative expressions are abbreviated as LN, LA, SDW, GPN, GPW, GPY, PPrc, PCC, PFC, PNC, PPC, PKC, PNO_3_^−^C, PNRA, NO_3_−, Chl, Car, LNC, LPC, LKC, LFeC, LMnC, LZnC, MDA, H_2_O_2_, O_2_^•^−, SSC, SPC, AsA, GSH, αToC, IAA, GA_3_, CK, Pro, ABA, Fv/Fm, PI, RWC, MSI, EL, SOD, CAT, APX, GR, *PvGR*, *PvSOD*, *PvCAT*, and *PvAPX*, respectively.

**Figure 14 plants-12-03435-f014:**
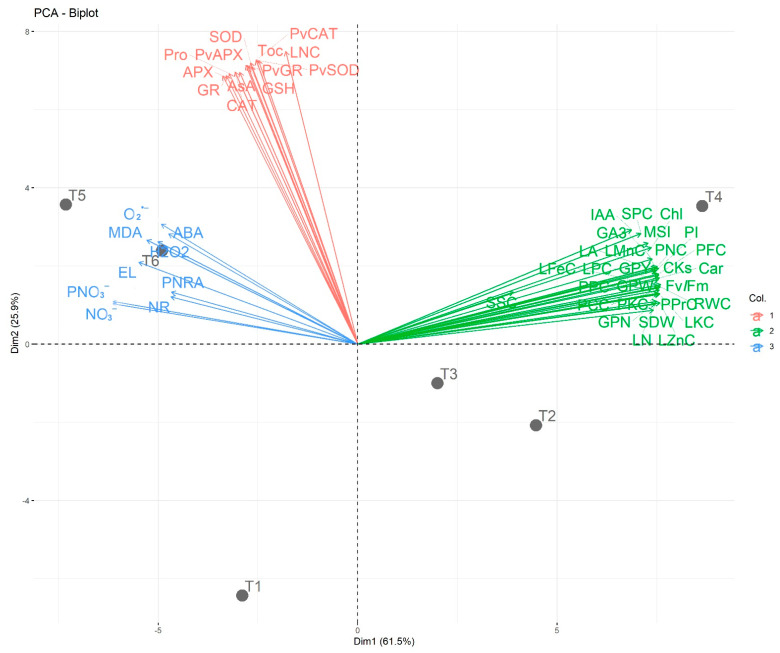
A biplot graph of examined treatments and parameters, revealing the two PCAs; Dim1 and Dim2 in *P. vulgaris* plants affected by the foliar supplementation of BHS under normal and excessive nitrogenous conditions. Col., color of clustered variables (based on K-means clustering); T1, 100%RND_0.0%BHS; T2, 100%RND_1.5%BHS; T3, 125%RND_0.0%BHS; T4, 125%RND_1.5%BHS; T5, 150%RND_0.0%BHS; and T6, 150%RND_1.5%BHS. The parameters number of leaves, leaves area, shoot dry weight, number of pods, pod weight, pod yield, pod protein, carbohydrate, fiber, nitrogen, phosphorous, potassium, and nitrate contents, pod nitrate reductase activity, leaf nitrate, chlorophyll, carotenoid, nitrogen, phosphorus, potassium, iron, manganese, zinc, malondialdehyde, hydrogen peroxide, superoxide, soluble sugar, soluble protein, ascorbate, glutathione, α-tocopherol, indole-3-acetic acid, gibberellic acid, cytokinins, proline, and abscisic acid contents, photosystem II quantum efficiency, performance index, relative water content, membrane stability index, electrolyte leakage, superoxide dismutase, catalase, ascorbate peroxidase, and glutathione reductase activities, and *GR*, *SOD*, *CAT*, and *APX* relative expressions are abbreviated to LN, LA, SDW, GPN, GPW, GPY, PPrc, PCC, PFC, PNC, PPC, PKC, PNO_3_^−^C, PNRA, NO_3_^−^, Chl, Car, LNC, LPC, LKC, LFeC, LMnC, LZnC, MDA, H_2_O_2_, O_2_^•−^, SSC, SPC, AsA, GSH, αToC, IAA, GA_3_, CK, Pro, ABA, Fv/Fm, PI, RWC, MSI, EL, SOD, CAT, APX, GR, *PvGR*, *PvSOD*, *PvCAT*, and *PvAPX*, respectively.

**Table 1 plants-12-03435-t001:** Fresh raw clover honey analysis.

Component.	Unit	Value	Component		Unit	Value
Proteins	%	0.30 ± 0.01	Osmoprotectants:			
Organic acids		0.51 ± 0.02	Proline		mg kg^−1^ FW	50.1 ± 2.04
pH		4.02 ± 0.14	Total soluble sugars		%	81.9 ± 2.42
Minerals:			Antioxidants and Vitamins:			
K	mg kg^−1^ FW	460 ± 11.2	Vitamin	C	mg kg^−1^ FW	30.1 ± 1.02
P		49.8 ± 1.85		B1		0.15 ± 0.00
Mg		83.6 ± 2.61		B2		0.19 ± 0.00
Ca		70.2 ± 1.92		B3		1.70 ± 0.07
Fe		69.5 ± 1.84		B5		1.05 ± 0.05
Mn		8.62 ± 0.30		B6		2.21 ± 0.11
Zn		5.64 ± 0.18		B9		0.22 ± 0.01
Cu		4.58 ± 0.15	DPPH radical-scavenging activity		%	89.5 ± 3.01
Iodine		80.8 ± 2.44				
Selenium		0.94 ± 0.04				

## Data Availability

The data presented in this study are available upon request from the corresponding author.

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
