# Peer review of "Effect of Eco-Friendly Application of Bee Honey Solution on Yield, Physio-Chemical, Antioxidants, and Enzyme Gene Expressions in Excessive Nitrogen-Stressed Common Bean (Phaseolus vulgaris L.) Plants"

_plants, 2023, doi:10.3390/plants12193435_

Round 1

Author Response

Plants - MDPI

Manuscript ID: Plants-2471930

Manuscript Title:  "Eco-Friendly Application of Bee-Honey Solution on Yield, Physio-Chemical, Antioxidants, and Enzyme Gene Expressions in Excessive N-Stressed Phaseolus vulgaris Plants"

============================================================

Dear Ms. Lilia Li

Assistant Editor, Plants

         Thank you for your efforts and we would also like to thank the reviewers a lot for their valuable comments that helped improve our manuscript. We have corrected the manuscript based on the reviewers' comments, corrections made in the text in brown, and outlined step by step as follows:

Response to the comments of Reviewer#1:

The work is well written. In the introduction, the authors justified the selection of the research topic in detail. The chapter Material and Methods describes the preparation of the research material, while references to authors' publications are made regarding the research methods used in the work. Results and discussion: the results obtained are presented in the form of 1 table and 14 figures and Supplementary Materials. The obtained data were thoroughly compared with the data of other researchers. The conclusions are consistent with the evidence and arguments presented. The conclusions respond to the set goal of the work.

Many thanks to Reviewer#1 for his positive comments

Mostafa M. Rady (Corresponding author)

Reviewer 2 Report

(1) The article is about eco-friendly utilization of bee-honey solution in yield, antioxidants, physio-chemical and enzyme gene expressions in excessive N-stressed bean plants. The main goal of this research was about the function of diluted bee-honey solution in decreasing the negative impacts of nitrogen toxicity of faba bean,yield quality and defense systems.

(2) Yes, the topic of the article is ORIGINAL, and the article can be very useful for researchers and scholars in this field.

(3) The article talks about the transcript levels of PvCAT, PvAPX,PvSOD, and PvGR, and the potential function of DHS as an eco-friendly knowledge in sustainable agriculture.

(4) Material and Methods has written very well, and it does not need any significant changes.

(5) The conclusion has written very well, and it has given the brief of the manuscript to authors with new directions, the good point of Conclusion is that, it does not Copy from other parts of the manuscript, and it does not similar with Abstract.

(6) Authors should re-check the format of all references, and also articles also need DOI which should be added to them. Also, all the references should double-checked with the references which have been used in the text.

(7) Tables and Figures are OK. The article just need Minor revision.

Reviewer 3 Report

Journal

Plants (ISSN 2223-7747)

Manuscript ID

plants-2603072

Type

Article

Title

Eco-Friendly Application of Bee-Honey Solution on Yield, Physio-Chemical, Antioxidants, and Enzyme Gene Expressions in Excessive N-Stressed Phaseolus vulgaris Plants

Authors

Hussein E.E. Belal , Mostafa A.M. Abdelpary , El-Sayed M. Desoky , Esmat F. Ali , Najla A.T. Al Kashgry , Mostafa M. Rady * , Wael M. Semida , Amr E. M. Mahmoud , Ali A.S. Sayed

Section

Crop Physiology and Crop Production

Special Issue

Legumes and Stressful Conditions

Bee-honey solution (BHS) is considered a plant growth multi-biostimulator because it is rich in osmoprotectants, antioxidants, vitamins, and mineral nutrients that can promote abiotic stress resistance in bean plants. 

The present research manuscript topic is investigated in the literature, and there is a very few of reference published. However, this paper gives significant contribution to the current knowledge in related field. The data are sound and it deserves to be published, after minor revisions.

Overall Recommendation: Minor Revisions

Prepare response letter with point-to-point replies to each comment.

Don't forget to highlight changes you want to made in the current version (attached).

Abstract:

1)      An abstract should be concise self-contained summary, should include the background/ objective, purpose of the study (including its statistical significance), methods, results and conclusion in one paragraph. Need improvements.

2)      Define abbreviation at first appearance and subsequently use abbreviated form.

Keywords:

3)      Keywords should not be the same as mentioned in the title or abstract.

Introduction

4)      Authors completely failed to develop the hypothesis with reference to title and objective, in the introduction section.

5)      Use www.turnitin.com to find and eliminate unnecessary self-repetition and any copied text.

6)      Don’t use introduction as second window of review of literature.

7)      Run spell check and proof the manuscript to avoid spelling mistakes and grammatical errors.

Materials and methods

8)      The text has many typing and grammatical errors, capitalization issues. English style and language requires a profound revision. However, the readability of the manuscript needs to be improved, preferably carefully reviewing by a native English speaker. All proper nouns must be abbreviated. Abbreviations must be described completely at first mention with brackets. Don’t start a sentence with an abbreviation here.

9)      Please add details for analytical methodologies to make it reproducible.

10)  Quality assurance of data is mandatory!!! How many batch, repeats, chemical grade and for used instruments manufacturers’ user manual and instructions were strictly followed or not!!!

Results and Discussion

11)  Data is sound one. It deserves to be published.

12)  Discussion needs improvements. Please cite Figure No. or Table No. in brackets at suitable places for a better connectivity in results and discussion sections as to facilitate the reader. I would have expected slightly greater discussion; more detail on the mechanisms and logical reasoning is required. There is much more scope here for discussing the implications of what the results means. Add suitable references throughout discussion section, where required.

Conclusion

13)  Novelty of this research work is again questionable with reference to practical significance and economic feasibility must be worked and mentioned. Kindly revise the conclusion subjected to conclusive findings only.

References

1)      A few very old references have been used. These must be updated with recent research findings or removed. Proper formatting is questionable. It must be according to MDPI plants Journal. References formatting are inconsistent. Verify each reference from original source and cross check references in the text and reference section.

1)      The text has many typing and grammatical errors, capitalization issues. English style and language requires a profound revision. However, the readability of the manuscript needs to be improved, preferably carefully reviewing by a native English speaker. All proper nouns must be abbreviated. Abbreviations must be described completely at first mention with brackets. Don’t start a sentence with an abbreviation here.

Author Response

Please the attached file.

Round 2

Reviewer 2 Report

1- Please, add the English name of Phaseolus vulgaris in TITLE.

2- Write and Arrange the keywords according to the alphabetic order.

3- In introduction section, please add the English name of Phaseolus vulgaris L. which is common bean before the latin name and put the scientific name of common bean inside the parentheses.

4- Introduction and Materials and Methods are written very well, and they do not need any changes.

5- Line 172, What is the usage of  (   ), in this part, please, delete (  ), and just write references 28,31,34.

6- This also should be done for Line 204... (28,31,34)

7- References are OK and enough.

Author Response

Plants - MDPI

Manuscript ID: Plants-2471930

Manuscript Title:  "Eco-Friendly Application of Bee-Honey Solution on Yield, Physio-Chemical, Antioxidants, and Enzyme Gene Expressions in Excessive N-Stressed Phaseolus vulgaris Plants"

==================================================

Dear Ms. Lilia Li

Assistant Editor, Plants

         Thank you for your efforts and we would also like to thank the reviewers a lot for their valuable comments that helped improve our manuscript. We have corrected the manuscript based on the reviewers' comments, corrections made in the text in red, and outlined step by step as follows:

Response to the comments of Reviewer#2:

1- Please, add the English name of Phaseolus vulgaris in TITLE.

Re: The English name “common bran” has been added before the scientific name, which in turn has been enclosed in parentheses (Line 4).

2- Write and Arrange the keywords according to the alphabetic order.

Re: The keywords have been arranged according to the alphabetic order (Line 35).

3- In introduction section, please add the English name of Phaseolus vulgaris L. which is common bean before the latin name and put the scientific name of common bean inside the parentheses.

Re: The English name “common bran” has been added before the scientific name, which in turn has been enclosed in parentheses (Line 38).

4- Introduction and Materials and Methods are written very well, and they do not need any changes.

Re: Many thanks to the reviewer for this positive comment.

5- Line 172, What is the usage of  (   ), in this part, please, delete (  ), and just write references 28,31,34.

Re: The brackets have been deleted (Line 173).

6- This also should be done for Line 204... (28,31,34)

Re: The brackets have been deleted (Line 205).

7- References are OK and enough.

Re: Many thanks to the reviewer for this positive comment.

Again, many thanks to Reviewer#2 for his valuable comments.

Mostafa M. Rady (Corresponding author)

Reviewer 3 Report

Significant improvements has been made.

Author Response

Plants - MDPI

Manuscript ID: Plants-2471930

Manuscript Title:  "Eco-Friendly Application of Bee-Honey Solution on Yield, Physio-Chemical, Antioxidants, and Enzyme Gene Expressions in Excessive N-Stressed Phaseolus vulgaris Plants"

============================================================

Dear Ms. Lilia Li

Assistant Editor, Plants

         Thank you for your efforts and we would also like to thank the reviewers a lot for their valuable comments that helped improve our manuscript. We have corrected the manuscript based on the reviewers' comments, corrections made in the text in red, and outlined step by step as follows:

Response to the comments of Reviewer#3:

Significant improvements have been made.

Many thanks to Reviewer#3 for his valuable comments that help us improve our manuscript.

Mostafa M. Rady (Corresponding author)